# Lymphocyte Subpopulations Associated with Neutralizing Antibody Levels of SARS-CoV-2 for COVID-19 Vaccination

**DOI:** 10.3390/vaccines10091550

**Published:** 2022-09-17

**Authors:** Wan-Ting Huang, Shao-Wen Weng, Hong-Tai Tzeng, Feng-Chun Yen, Yu-Shao Chiang, Huey-Ling You

**Affiliations:** 1Department of Laboratory Medicine, Kaohsiung Chang Gung Memorial Hospital, Kaohsiung 83301, Taiwan; 2Department of Pathology, Kaohsiung Chang Gung Memorial Hospital, Kaohsiung 833031, Taiwan; 3School of Medicine, College of Medicine, Chang Gung University, Taoyuan 33382, Taiwan; 4Department of Medical Laboratory Sciences and Biotechnology, Fooyin University, Kaohsiung 83102, Taiwan; 5Department of Internal Medicine, Kaohsiung Chang Gung Memorial Hospital, Kaohsiung 83301, Taiwan; 6Institute for Translational Research in Biomedicine, Kaohsiung Chang Gung Memorial Hospital, Kaohsiung 83301, Taiwan

**Keywords:** SARS-CoV-2 infection, COVID-19, vaccines, lymphocyte subpopulations, cytokines

## Abstract

The comprehensive knowledge regarding the immune response during coronavirus disease 2019 (COVID-19) vaccination is limited. The aim of this study was to longitudinally investigate not only the dynamic changes of peripheral lymphocyte subpopulations and cytokine levels but parallel changes of antibody levels against severe acute respiratory syndrome coronavirus 2 (SARS-CoV-2). Blood samples of 20 healthcare workers with two doses of COVID-19 vaccine were prospectively collected. The percentages of lymphocyte subpopulations from peripheral blood and cytokine production in lymphocytes with in vitro stimulation were assessed using eight-color flow cytometry. SARS-CoV-2 spike antibodies (anti-S Abs) and functional neutralizing antibodies (nAbs) were also measured. The relation between pre- and post-vaccination immunity was analyzed. There are 7 men and 13 women with a median age of 44.0 years (range: 25.7–59.5 years). The individuals had an increased percentage of lymphocytes at post-vaccination with statistical significance post first dose (*p* = 0.031). The levels of transitional cells (*p* = 0.001), such as plasmablasts (*p* < 0.001) and plasma cells (*p* = 0.031), were increased compared with pre-vaccination. Recent thymic emigrants of CD4+ T cells subsets were significantly higher at post-vaccination than those at pre-vaccination (*p* = 0.029). Intracellular levels of tumor necrosis factor-alpha, interferon-γ, interleukin (IL)-2, IL-21, transforming growth factor-beta and IL-17 produced by CD4+ T, CD8+ T, and natural killer cells were increased. All individual samples showed reactivity to anti-S Abs and the levels of nAbs were elevated after vaccination. The magnitude of adaptive immunity was associated with vaccine types and doses. Alterations of total memory B cells (*p* < 0.001), non-switched memory B cells (*p* = 0.016), and memory Treg cells (*p* < 0.001) were independent predictors for nAb levels. These findings might be helpful in elucidating the immune response of COVID-19 vaccination and in developing new strategies for immunization.

## 1. Introduction

The coronavirus disease 2019 (COVID-19) has rapidly spread since late December 2019. Along with the outbreak, novel vaccines clear the threshold for emergency use authorization by the U.S. Food and Drug Administration. Vaccine-elicited humoral and cell-mediated immunity effectiveness against severe acute respiratory syndrome coronavirus 2 (SARS-CoV-2) have been demonstrated [1,2,3,4]. The production of anti-SARS-CoV-2 antibodies and the generation of antigen-specific memory B cells (MBCs) can provide rapid serological immunity and evoke recalling responses, respectively [5,6]. SARS-CoV-2 specific T cells can be induced in the first 2 weeks after onset of the symptoms [7]. Robust T-cell responses enhance long-lasting immunity against reinfections [8,9].

Alteration of peripheral lymphocytes is associated with clinical response of COVID-19. In fact, lymphopenia is commonly observed during infection and is also associated with disease severity [10]. The longitudinal studies highlight dynamic changes of lymphocyte subsets in the course of COVID-19. Deng et al. reported a downward to stable trend of B-, T and NK- cells in the first two weeks and gradually increased to normal levels in the fifth week in non-severe patients [11]. Similarly, Huang et al. also demonstrated the lowest T-lymphocyte count on the 14th day followed by returning to the normal level in the improved patients or remaining low in the unimproved patients [12]. Additionally, T cells had activated phenotypes with increased HLA DR+ and CD38+ subsets during SARS-CoV-2 infection [13]. Regulator T (Treg) cells, essential for immune homeostasis, are reduced, especially in severe COVID-19 [14,15]. These findings indicate hyperactivation and dysregulation of immune systems in patients with SARS-CoV-2 infection.

The immunity induced by COVID-19 vaccines may increase anti-SARS-CoV-2 antibodies with expansion of specific MBCs and plasmablasts [2,6]. The magnitude of vaccine-induced B-cell immunity is higher in the individuals with a history of SARS-CoV-2 infection than those without it [16,17]. In addition, Pape et al. reported better antigen-binding capacity of MBCs in individuals previously infected than those vaccinated alone [17]. However, a comprehensive analysis of peripheral lymphocyte subpopulations and cytokine profiles in vaccinated health individuals is limited. Previously, we had conducted a flow cytometry study to differentiate the subsets and cytokine profiles of the lymphocyte populations [18]. The potential of lymphocyte production, differentiation and activation, and cytokine production may reflect the immune status of stimulated individuals.

In this study, we aimed to describe alterations of lymphocyte subsets and cytokine levels, and also to delineate the dynamics of neutralizing antibody levels (nAbs) against SARS-CoV-2 during COVID-19 vaccination. To this end, we analyzed the most relevant T, B, and natural killer (NK) cell subpopulations and intracellular cytokines using eight-color flow cytometry. SARS-CoV-2 spike antibodies (anti-S Abs) and nAbs were measured. Comprehensive analysis of lymphocyte subsets, cytokines and antibody levels may clarify the immune response to SARS-CoV-2 vaccines.

## 2. Materials and Methods

### 2.1. Study Population and Sample Collection

This study was a prospective study and approved by the Chang Gung Memorial Hospital Ethical Committee (201901509B0 and 202101124B0). From July 2021 to June 2022, healthy medical staff with baseline data of lymphocyte subsets and cytokine secretions, tested in the previous study were enrolled [18]. Written informed consent was obtained from all participants. Whole blood samples were collected in ethylenediaminetetraacetic acid tube (Becton Dickinson, NJ, USA) before and after vaccine administration. Total white blood cells (WBC) and percentages of lymphocytes were assayed using an automated hematology analyzer Sysmex XN-9000 (Sysmex, Kobe, Japan), followed by collecting plasma and peripheral blood mononuclear cells (PBMCs) through phase separation of plasma layer and Buffy coat phase, respectively. PBMCs were prepared for flow cytometry analysis of lymphocyte subsets and intracellular cytokines. Plasma were aliquoted and stored at −80 °C for further COVID-19 antibody tests.

### 2.2. SARS-CoV-2 Antibody Detection Assay

SARS-CoV-2 spike antibodies (anti-S Abs) was measured by the Elecsys anti-SARS-CoV-2 assay (Roche Diagnostics International Ltd., Rotkreuz, Switzerland) performed on Roche Cobas e801 (Roche Diagnostics). Functional nAbs were evaluated using the SARS-CoV-2 surrogate virus neutralization test kit according to the manufacturer’s instruction (Cat# L00847-A, GenScript, Netherlands). Briefly, serum samples as well as positive and negative assay controls were diluted 1:10 in sample dilution buffer and then mixed with an equal volume of horseradish peroxidase-conjugated receptor binding domain (RBD). Both controls and samples were tested in duplicates. After incubation for 30 min at 37 °C, 100 µL of mixture was transferred to a 96-well plate pre-coated with recombinant angiotensin-converting enzyme 2. After incubation at 37 °C for 15 min, the supernatant was removed and 100 µL of tetramethylbenzidine substrate was added and incubated for further 15 min at room temperature. Finally, the reaction was stopped by addition of 50 µL stop solution. The absorbance of the resulting solution was measured spectrophotometrically at 450 nm (Sunrise, TECAN, Switzerland). The average optical density (OD) of each individual samples was used to calculate the inhibition %. The formula was as follows:Inhibition = [ (1 − (OD value of Sample/OD of Negative Control)] × 100%.

### 2.3. Lymphocyte Subpopulations

We analyzed lymphocytes subsets in the PB using eight-color flow cytometric analysis as previously reported (Appendix A) [19]. T cells were further divided into (1) recent thymic emigrants (RTEs, CD45RA+CD62L+CD31+); (2) naïve (CCR7+CD45RA+CD45RO−); (3) central memory (TCM, CCR7+CD45RA−CD45RO+); (4) effector (CCR7−CD45RA+CD45RO−); (5) effector memory (TEM, CCR7−CD45RA−CD45RO+); (6) activated (CD38+HLA-DR+); (7) and CC chemokine receptor (CCR)-associated (CCR3, CCR5, or CCR6) subsets. The regulatory T-cells (Treg, CD4+CD25+CD127low) were also be divided into naïve (CD45RO−), memory (CD45RO+), and activated (HLA-DR+) subsets. The combination of CD21 and CD27 allowed to divide CD19+ B cells into (1) naïve (CD21+CD27−); (2) memory (CD21+CD27+); and (3) CD21low subpopulations. Among total MBCs, non-switched and switched subsets were differentiated by IgM versus IgD delineation. Based on IgM expression, transitional B-cells (IgM+++CD38+++) and plasmablasts (IgM−CD38+++) were separated. Plasma cells were defined by the CD27+CD138+ phenotype. For NK-cell subsets, lymphocytes were first separated either by CD3−CD161+CD122+ or CD3−CD16+/CD56+ phenotypes. Then CD16 versus CD56 expression plots were divided into CD56+ bright, CD56+CD16− and CD56+CD16+ subsets. The expression levels of NK group 2 D/CD94 were determined.

### 2.4. Cytokine Secretion Assay

The cytokine secretion ability of B, T, and NK cells was detected using flow cytometry as previously reported (Appendix A) [18]. The response of intracellular cytokines after vaccination was evaluated through stimulation with or without the leukocyte activation cocktail (BD GolgiPlug™, BD Biosciences, Heidelberg, Germany). The levels of intracellular cytokines after stimulation were quantitatively determined by comparing the baseline of lymphocytes without treatment. We used a FACS Canto II flow cytometer (BD Biosciences, Heidelberg, Germany) configured with three lasers (405 nm violet laser, 488 nm blue laser, and 647 nm red laser) for data acquisition and the FACS DIVA software (BD Biosciences) for data analysis.

### 2.5. Statistical Analysis

All statistical analyses were performed using SPSS for Windows 11.0 software (SPSS Inc., Chicago, IL, USA). Comparisons between groups were performed using the Friedman test with pair-wise comparisons for three dependent samples, Wilcoxon signed rank test for two dependent samples, and Mann–Whitney U test for two independent samples. We used the generalized estimation equation (GEE) for analysis to account for within-subject correlations that present in longitudinal data across two vaccine doses. In GEE, nAbs was the response variable, whereas vaccination types and alterations of lymphocyte subsets were the main covariates. Other variables included as covariates were age, sex, WBCs and lymphocytes (%). All reported *p* values are from two-tailed tests, and *p* values of 0.05 or less were considered to indicate statistical significance.

## 3. Results

### 3.1. Baseline Characteristics

A total of 20 healthy medical staff were enrolled into this study. There are 7 men and 13 women with a median age of 44.0 years (range: 25.7–59.5 years). In the cohort, 13 of (65.0%) 20 individuals received two doses of the ChAdOx1-S vaccine at an interval of 10–11 weeks and 5 (25.0%) received two doses of the mRNA-1273 vaccine at an interval of 5 weeks. The other two individuals received first dose of the ChAdOx1-Svaccine and second dose of the mRNA-1273 vaccine at the interval of 12 weeks. The average days of blood collection after being vaccinated were 15.0 days (range: 13.0–18.0 days) after first dose and 36.7 days (range: 37.0–40.0 days) after second dose.

### 3.2. Alterations of Lymphocyte Subsets in the Course of COVID-19 Vaccination

We compared percentages of WBC and lymphocyte subsets at three time points, pre-vaccination, post first dose and post second dose. Table 1 showed the distribution and significance of their differences in detail. Representative data were illustrated in Figure 1. The individuals had an increased percentage of lymphocytes at post-vaccination with statistical significance at post first dose (*p* = 0.031). The levels of IgM+++CD38+++ transitional cells (*p* = 0.001), plasmablasts (*p* < 0.001) and plasma cells (*p* = 0.031) were above the baseline. In CD4+ T cells, RTEs (CD45RA+CD62L+CD31+), naïve (CD45RA+CD45RO−CCR7+), TEM (CD45RA−CD45RO+CCR7−) and activated (CD38+HLA−DR+) subsets were significantly higher at post-vaccination than at pre-vaccination. A similar trend was noted in CD8+ T cell subsets with the significantly higher frequency of CD38+HLA-DR+ and TCM (CD45RA−CD45RO+CCR7+) subpopulations. The overall percentage of Treg cells was (*p* < 0.001) decreased, but the frequency of memory Treg subsets was increased after second dose vaccination, suggesting the dynamic changes of Treg subpopulations shifting toward the memory phenotype.

### 3.3. Lymphocyte Response to COVID-19 Vaccine Brands

In order to determine the kinetic changes of lymphocyte subsets to the vaccine types and doses, we further analyzed the difference between groups with lymphocyte levels of pre-vaccination as the baseline. In the context of B cells, the individuals with the mRNA-1273 vaccine had the increased frequency in total MBCs (median: 0.8, interquartile range [IQR]: −0.75 to 9.8 vs. median: −2.9, IQR: −7.6 to −0.9, *p* = 0.011) and non-switched MBCs (median: 0.8, IQR: −0.5 to 3.8, vs. median: −2.2, IQR: −3.9 to −1.5, *p* = 0.002) than those with the adenovirus-based ChAdOx1-S vaccine after first dose of vaccination. For Treg cells, the percentages showed a trend of decline after vaccination (Table 2). However, the percentage of memory Treg cells (CD45RO+) gradually increased. The individuals with the mRNA-1273 vaccine had a significant higher frequency of memory Treg cells than those with the ChAdOx1-S-vaccinated cohort after first dose of vaccination. An inverse trend was observed for naïve Treg cells (CD45RO−). Although the levels of lymphocyte percentages declined in the individuals with the mRNA-1273 vaccine, total lymphocyte counts showed no remarkable change. Other lymphocyte subsets showed no significant difference between brands.

### 3.4. Cytokines Alterations and Response to COVID-19 Vaccine Brands

We then analyzed the response of intracellular cytokines to vaccination after stimulation with a polyclonal cell activation mixture containing the phorbol 12-myristate 13-acetate and ionomycin, and the brefeldin A was used to block the extracellular release of cytokines. Significant differences were observed between the pre- and post-vaccination with increased levels of all cytokines in CD4+ and CD8+ T cells, except interleukin (IL)-2, IL-21, transforming growth factor-beta (TGF-β) and IL-17 for NK cells (Figure 2). When we stratified the individuals by the vaccine types, the difference was associated with the vaccine dose (Table 3). For CD4+ T cells, tumor necrosis factor-alpha (TNF-α) (*p* < 0.001), interferon-γ (IFN-γ) (*p* = 0.002), IL-2 (*p* = 0.002) and IL-21 (*p* < 0.001) remained at high levels in the individuals with the mRNA-1273 vaccine after second dose. A similar trend was observed in CD8+ T cells. Expression of IL-17 in T cells was relatively evident in the individuals vaccinated by the mRNA-1273 vaccine. Meanwhile, the percentages of TNF-α-producing NK cells (*p* = 0.028) and IFN-γ-producing NK cells (*p* = 0.010) were significantly higher in the individuals with the mRNA-1273 vaccine compared to those with the ChAdOx1-S vaccine after second dose.

### 3.5. Longitudinal Predictors for Determination of COVID-19 nAbs

To determine the relationship between lymphocyte subsets and antibodies for SARS-CoV-2 virus, levels of anti-S Abs and nAbs were measured first. All individuals tested showed reactivity to anti-S Abs and elevation of nAbs after vaccination. High levels of antibodies were produced following the second dose of vaccination (Figure 3). Response was significantly higher and more rapid in those vaccinated subjects by the mRNA-1273 than by the ChAdOx1-S (Table 2). We then analyzed longitudinally for lymphocyte subsets for prediction of nAbs. In addition to the vaccine types (*p* = 0.002), increase in the percentages of total MBCs (*p* < 0.001), non-switched MBCs (*p* = 0.016), and memory Treg cells (*p* < 0.001) were independent predictors for nAb levels, adjusting the factors of age, sex, WBCs and lymphocyte percentages (Table 4, Figure 4).

## 4. Discussion

To establish effective immune response against SARS-CoV-2 requires participation and interaction of lymphocytes. In terms of lymphocyte subsets and cytokine levels, the differences between COVID-19 patients have been explored. The majority of SARS-CoV-2-specific T cells have the central and effector memory phenotype and are significantly related to disease severity [7,20]. These SARS-CoV-2-specific T cells represent a part of the adaptive immune system and regulate inflammation through type 1 interferon or effector cytokines (IFN-γ, TNF-α and IL-2) for an effective immune response [7,21]. However, increased levels of type 2 cytokines such as IL-4, IL-5, IL-13, IL-9 and IL-10 may impair anti-viral response and associate with poor prognosis [21].

In our study, we clarified the immune response after vaccination. The lymphocytes were not only activated but expanded. The frequency of transitional B cells was increased, which are referred to as recent bone marrow emigrants and subjected to a selection process against autoreactivity [22]. Likewise, the percentages of RTEs and naïve T cells were increased, suggesting the replenishment and diversity of T cells in the PB after vaccination [23]. Our results might indicate a possible high output of both bone marrow and thymus upon vaccine-induced immunity. This would promote a positive effect on regeneration of B and T cells with a broad repertoire for further sharpening of antigen recognition. In addition, comprehensive statistical analysis highlighted immune differences between the groups. The GEE used for modeling longitudinal data produced reasonable estimates of model parameters. Vaccine types, total MBCs, non-switched MBCs, and memory Treg were independent predictors for nAb levels.

Specific humoral immunity against SARS-CoV-2 plays important roles in protection for viral infection. We found that the percentages of MBCs and non-switched MBCs were significantly increased in the individuals vaccinated by mRNA-1273 and positively correlated with production of nAbs. For mRNA vaccines, anti-S IgG and anti-RBD antibodies reach the highest levels around 7 days after the second dose of vaccination and display progressive and significant decline after 3 months [5,6,17]. In contrast to the decline of serum antibodies, spike-specific MBCs continue expand for at least 9 months after the second dose of vaccination [5,6]. Vaccine-induced immune memory may effectively limit breakthrough COVID-19 infections by the rapid increase of serum antibodies and expansion of specific MBCs and plasmablasts [2,6]. Sokal et al. demonstrated mRNA vaccination effectively elicited RBD-specific MBCs with high affinity toward SARS-CoV-2 variants [24]. The affinity maturation of MBCs may go on for weeks, which continually produce spike-specific antibodies with somatic hypermutation [25,26]. Turner et al. study further highlighted persistent responses of germinal centers in lymph nodes for at least 12 weeks after the booster immunization [16]. Individuals receiving three doses of an mRNA vaccine may have expansion of memory B cell clones through a germinal center-independent process [27].

Peripheral CD4+ T cell subsets are correlated with COVID-19 severity. Gong et al. reported increased frequencies of TEM and EM T follicular helper (Tfh-EM) cells in convalescent patients, with the strongest magnitude in those recovering from severe COVID-19 [28]. The Tfh-EM cells may positively regulate antibody protection against SARS-CoV-2. Follicular Treg cells, which constrain antibody response, were reduced. Likewise, Painter et al. study showed vaccine-induced CD4+ T cells predominately exhibiting phenotypes of T helper 1 and Tfh cells and correlating with humoral and cellular responses [29]. Meckiff et al. further refined SARS-CoV-2-reactive Tfh cells using single-cell transcriptomic analysis [30]. The cytotoxic Tfh cells were clarified, which were associated with impairment of B cell immune responses to SARS-CoV-2 and a decrease of anti-S1/S2 antibody levels in hospitalized COVID-19 patients. The response of cytotoxic Tfh cells could be reduced by follicular Treg cells [30,31]. However, the generation of SARS-CoV-2-reactive Treg cells might be decreased in hospitalized COVID-19 patients [30]. Thus, Treg cells may modulate immune responses against COVID-19. One key observation from our data is the alteration of CD4+ TEM and Treg cells post vaccination, with trends in frequency similar to patients infected by SARS-CoV-2 [14,15]. In contrast, the proportion of memory Treg cells was increased and positively associated with the magnitude of nAbs. These findings indicate potential activation of Treg cells over COVID-19 vaccination might reduce the response of cytotoxic Tfh cells. However, what role of Treg cells in the effective production of nAbs is yet to be answered. Further studies are required to clarify the relationship.

Cytokines generated by lymphocytes are important against SARS-CoV-2 but contribute to cytokine release syndrome in patients with severe COVID-19 infection [32,33]. Coordinated cytokine network is necessary for effective immunization. The present study showed cytokines profiles of CD4+ T cells, CD8+ T cells, and NK cells upon vaccination. Compared to the baseline of individuals, the higher levels of TNF-α, IFN-γ, IL-2 and IL-21 may suggest activation of innate and adaptive immunity. The PBMCs responding to stimulation for producing predominant type 1 cytokines were higher in the individuals vaccinated by mRNA than adenovirus vector vaccines. This is consistent with Sahin et al. study that demonstrated mRNA vaccination inducing robust T helper type 1 immune responses [34]. The immune response induced by adenovirus vector vaccines declined early after the second dose, in contrast to the long lasting cellular immune response from mRNA vaccination [3].

## 5. Conclusions

In summary, we found the alterations of peripheral lymphocyte subpopulations with an increase in cytokine responses during COVID-19 vaccination. Both B and T cells were activated with increased bone marrow and thymic output, respectively. The percentage of Treg cells was decreased but displayed an activated phenotype. The magnitude of adaptive immunity was associated with vaccine types and doses, and mRNA vaccination induced robust T helper type 1 immune responses. Levels of nAb were positively correlated with alterations of total MBCs, non-switched MBCs and memory Treg cells. These findings might be helpful in elucidating the immune response of COVID-19 vaccination in developing new strategies for immunization.

## Figures and Tables

**Figure 1 vaccines-10-01550-f001:**
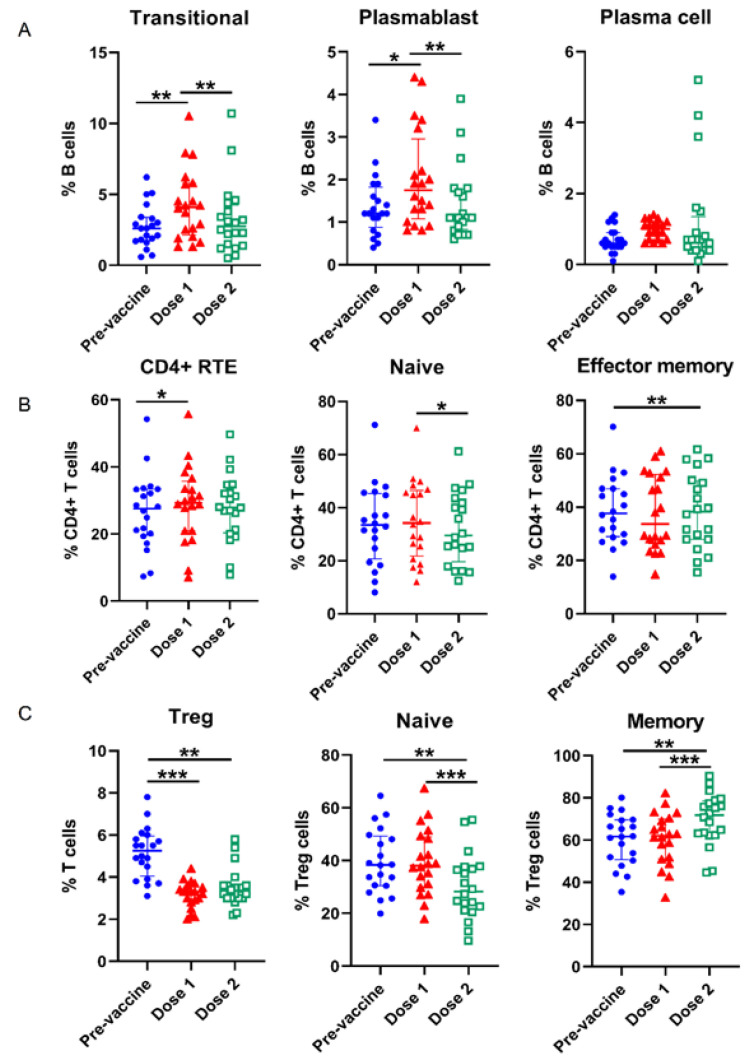
Changes of lymphocyte subpopulations between COVID-19 vaccine doses. (**A**): B, (**B**): CD4+ T, (**C**): Treg cells. Dot plots are representative of the percentages of lymphocyte subsets. The positions of the median and the quartiles are shown. “*” represents *p* < 0.05; “**” represents *p* < 0.01; “***” represents *p* < 0.001. RTE, recent thymic emigrants.

**Figure 2 vaccines-10-01550-f002:**
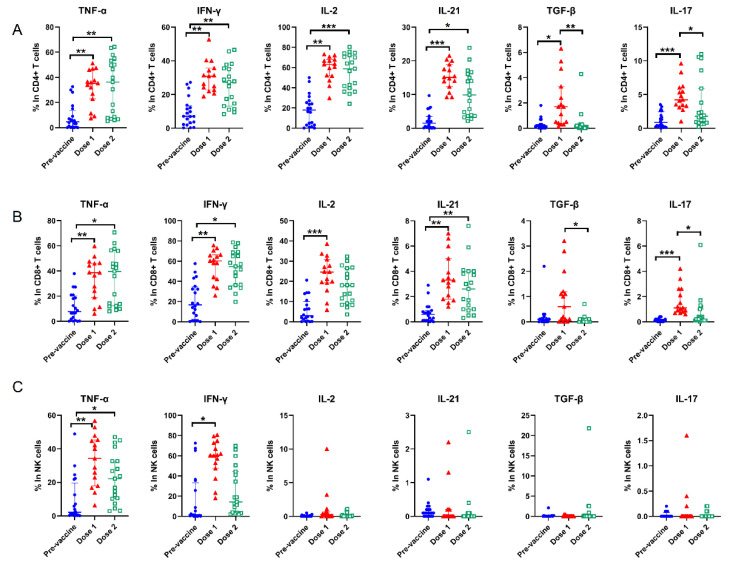
Changes of intracellular cytokines between COVID-19 vaccine doses. (**A**): CD4+ T, (**B**): CD8+ T, (**C**): NK cells. Dot plots represent the percentages of lymphocytes with intracellular cytokine production. The positions of the median and the quartiles are shown. “*” represents *p* < 0.05; “**” represents *p* < 0.01; “***” represents *p* < 0.001. Pre-vaccine, n = 20; Dose 1, n = 15; Dose 2, n = 19.

**Figure 3 vaccines-10-01550-f003:**
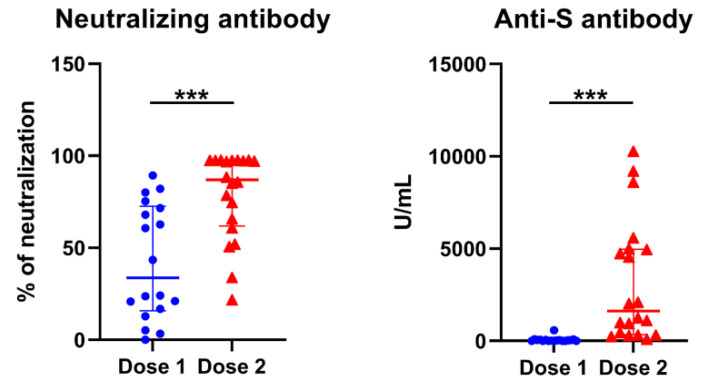
Comparison of SARS-CoV-2 neutralizing and anti- spike antibodies between COVID-19 vaccine doses. The positions of the median and the quartiles are shown “***” represents *p* < 0.001. Dose 1, n = 18; Dose 2, n = 20.

**Figure 4 vaccines-10-01550-f004:**
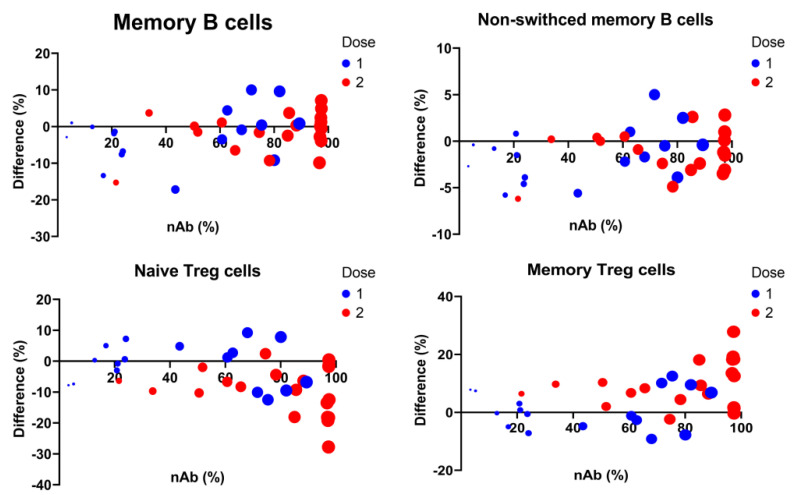
Relation between SARS-CoV-2 neutralizing antibody (nAb) levels and alterations of lymphocyte subpopulations. Difference represents changes of the lymphocyte percentages post-vaccination, compared to the baseline. Dot size indicates nAb levels.

**Table 1 vaccines-10-01550-t001:** Response of lymphocyte subpopulations to COVID-19 vaccination.

Variable	Vaccination			*p* Value
	Baseline	After 1st Dose	After 2nd Dose	
**White blood cells (×10^3^ cells/μL)**	6.4 (5.2, 7.4)	6.3 (5.6, 7.0)	6.3 (5.2, 7.5)	1.000
**Lymphocytes (%)**	31.9 (29.2, 35.2) ^a^	35.3 (30.5, 43.7) ^a^	35.0 (28.3, 42.9)	0.031
**T cells (% lymphocytes)**	64.4 (58.3, 72.4)	63.3 (56.0, 70.7)	63.5 (56.4, 73.5)	0.819
**α/β T cells (% T cells)**	89.8 (86.6, 93.0)	88.8 (85.6, 93.4)	87.4 (82.2, 92.0)	0.142
**γδ T cells (% T cells)**	9.4 (6.2, 12.5) ^b^	10.0 (5.6, 12.7)	8.9 (5.3, 12.6) ^b^	0.047
**CD4+ T cells (% T cells)**	33.2 (28.5, 38.7)	33.0 (26.9, 38.5)	32.2 (26.7, 38.3)	0.626
CD45RA+CD62L+CD31+ (% CD4+ cells)	26.7 (19.5, 33.4) ^a^	29.0 (20.9, 35.7) ^a^	27.9 (20.3, 34.0)	0.029
Naïve (% CD4+ cells)	33.6 (20.8, 45.4)	35.1 (21.8, 46.6) ^c^	32.2 (19.6, 43.2) ^c^	0.015
CD38+HLA-DR+ (% CD4+ cells)	4.2 (2.7, 4.6)	4.8 (3.7, 5.7) ^c^	4.1 (3.0, 4.8) ^c^	0.021
TCM (% CD4+ cells)	16.7 (13.5, 19.1)	15.6 (12.9, 17.8)	16.2 (11.4, 18.1)	0.109
TEM (% CD4+ cells)	38.8 (28.9, 46.9) ^b^	39.4 (28.3, 49.9)	40.9 (31.4, 50.3) ^b^	0.004
TE (% CD4+ cells)	6.0 (2.2, 8.3)	4.8 (2.9, 5.9)	5.1 (2.9, 6.4)	0.705
CCR5+ (% CD4+ cells)	18.5 (12.2, 27.5)	19.4 (13.5, 26.2)	19.7 (14.4, 23.2)	0.387
CCR3+ (% CD4+ cells)	3.6 (1.2, 3.6) ^ab^	1.9 (0.8, 2.4) ^a^	1.7 (0.7, 2.3) ^b^	0.006
CCR6+ (% CD4+ cells)	15.7 (12.1, 19.5)	18.9 (15.1, 22.7)	19.3 (15.2, 22.1)	0.127
**CD8+ T cells (% T cells)**	25.3 (19.3, 29.9)	24.7 (20.4, 28.1)	25.6 (20.6, 30.0)	0.182
CD45RA+CD62L+CD31+ (% CD8+ cells)	22.3 (12.7, 27.6)	25.4 (15.4, 34.0)	24.5 (16.1, 34.3)	0.182
Naïve (% CD8+ cells)	31.4 (15.7, 41.1)	33.6 (18.5, 50.9)	31.2 (18.8, 43.1)	0.064
CD38+HLA-DR+ (% CD8+ cells)	13.6 (9.0, 15.5) ^a^	16.2 (10.2, 18.8) ^ac^	14.3 (10.0, 19.1) ^c^	0.001
TCM (% CD8+ cells)	1.7 (1.1, 2.4) ^a^	1.3 (0.6, 1.6) ^a^	2.0 (0.8, 1.7)	0.004
TEM (% CD8+ cells)	38.8 (28.4, 53.9)	37.6 (26.3, 52.2)	38.3 (27.9, 49.9)	0.462
TE (% CD8+ cells)	21.7 (12.7, 26)	21.4 (11.6, 32.3)	22.2 (10.4, 31.1)	0.892
TEM CCR5+ (% CD8+ cells)	26.8 (16.9, 33.3)	25.5 (16.8, 37.0) ^c^	27.6 (17.4, 41.1) ^c^	0.022
TE CCR5+ (% CD8+ cells)	7.4 (3.3, 9.6)	8.6 (4.5, 9.9)	10.7 (4.1, 14.5)	0.158
**Regulatory T (Treg) cells (% T cells)**	5.2 (4.1, 6.0) ^ab^	3.2 (2.8, 3.6)^a^	3.5 (3.0, 3.6) ^b^	<0.001
Naïve (% Treg cells)	39.7 (30.5, 49.3) ^b^	39.4 (30.0, 49.0) ^c^	30.1 (21.5, 37.4) ^bc^	<0.001
Memory (% Treg cells)	60.4 (50.7, 69.6) ^b^	60.6 (51.1, 70.0) ^c^	69.9 (62.6, 78.5) ^bc^	<0.001
HLA-DR+ (% Treg cells)	14.6 (11.2, 18.6) ^b^	11.9 (8.3, 15.3)	11.5 (8.9, 13.7) ^b^	0.019
**B cells (% lymphocytes)**	9.5 (6.6, 12.2)	10.2 (7.8, 12.5)^c^	9.3 (7.4, 11.7) ^c^	0.032
Transitional (% B cells)	2.7 (1.7, 3.4) ^a^	4.3 (2.1, 5.8) ^ac^	3.3 (1.4, 4.3) ^c^	0.001
Naïve (% B cells)	69.3 (58.6, 82.5)	67.8 (52.7, 79.5)	69.7 (57.7, 78.7)	0.308
Non-switched memory (% B cells)	8.8 (4.9, 10.5)	7.2 (3.6, 11.2)	7.7 (3.8, 8.7)	0.091
Switched memory (% B cells)	11.6 (6.0, 14.6)	11.0 (6.8, 16.1)	11.4 (8.4, 14.5)	0.316
CD21^lowCD38^low (% B cells)	5.7 (2.3, 9.0)	7.3 (4.1, 10.0)	6.2 (2.7, 9.5)	0.165
Plasmablast (% B cells)	1.4 (0.9, 1.8) ^a^	2.0 (1.1, 3.0) ^ac^	1.4 (0.8–1.8) ^c^	<0.001
Plasma cell (% B cells)	0.7 (0.5, 0.9)	1.0 (0.7, 1.2)	1.2 (0.4, 1.4)	0.031
**NK cells (% lymphocytes)**	24.9 (19.8, 30.9)	25.6 (16.0, 33.2)	25.8 (17.2, 33.3)	0.729
CD56+ bright (% NK cells)	4.3 (2.1, 6.0)	3.8 (1.8, 5.2)	5.0 (2.4, 6.2)	0.115
CD56+CD16- (% NK cells)	3.3 (1.3, 4.5)	2.9 (1.6, 4.1) ^c^	4.1 (1.5, 5.4) ^c^	0.031
CD56+CD16+ (% NK cells)	94.2 (92.4, 96.2)	94.8 (93.5, 96.8)	93.6 (91.6, 96.5)	0.287
CD94/NKG2D (% NK cells)	50.3 (38.1, 67.2)	49.5 (39.2, 65.0)	50.0 (39.0, 66.5)	0.767

Data presented in median values (IQR). Small letters represent statistically significant differences among the groups (Friedman post-hoc Dunn test). NK, natural killer; NKG2D, NK group 2D; TCM, central memory T cell; TE, effector T cell; TEM, effector memory T cell.

**Table 2 vaccines-10-01550-t002:** Differences of regulatory T cells between ChAdOx1-S and mRNA-1273 vaccines.

	After First Dose			After Second Dose		
	ChAdOx1-S (n = 15)	mRNA-1273 (n = 5)	*p* Value	ChAdOx1-S (n = 13)	mRNA-1273 (n = 7)	*p* Value
Anti S Ab (U/mL)	26.9 (1.4, 56.2)	33.6 (8.4, 329.4)	0.661	937.0 (292.5, 1617.5)	5568.0 (4716.0, 9182.0)	<0.001
Neutral level (%)	23.6 (8.9, 61.7)	75.4 (46.2, 85.7)	0.026	78.3 (51.1, 92.6)	97.3 (96.7, 97.4)	0.037
White blood cells (×10^3^ cells/μL)	−0.4 (−0.74, 1.2)	−0.2 (−0.8, 1.5)	0.612	0.3 (−0.9, 0.6)	0.1 (−0.9, 0.9)	0.817
Lymphocytes (%)	4.1 (1.7, 8.8)	−1.2 (−4.8, 5.5)	0.138	6.8 (2.7, 8.9)	−1.8 (−6.1, 5.2)	0.036
Regulatory T (Treg) cells (% T cells)	−1.9 (−2.4, −1.4)	−2.6 (−3.3, −2.1)	0.066	−1.7 (−2.5, −1.5)	−1.6 (−2.4, −0.6)	0.393
Naïve (% Treg cells)	2.7 (−0.8, 7.8)	−9.5 (−11.3, −4.9)	0.002	−6.4 (−10.0, −1.9)	−13.5 (−18.3, −6.7)	0.097
Memory (% Treg cells)	−2.7 (−7.8, 0.8)	9.5 (4.9, 11.3)	0.002	6.4 (1.9, 10.0)	13.5 (6.7, 18.3)	0.097
HLA-DR+ (% Treg cells)	−3.0 (−7.9, −1.2)	0.4 (−6.2, 4.2)	0.349	−3.9 (−5.4, −1.4)	−3.6 (−0.3, 0.7)	0.275

Data compared to the baseline and presented in median values (interquartile range).

**Table 3 vaccines-10-01550-t003:** Differences of intracellular cytokine production between ChAdOx1-S and mRNA-1273 vaccines.

	After First Dose			After Second Dose		
	ChAdOx1-S (n = 10)	mRNA-1273 (n = 5)	*p* Value	ChAdOx1-S (n = 12)	mRNA-1273 (n = 7)	*p* Value
**CD4+ T cells**						
TNF-α	22.0 (16.2, 30.9)	34.6 (13.8, 41.2)	0.440	6.5 (3.6, 24.2)	46.5 (30.8, 57.4)	<0.001
IFN-γ	19.0 (11.9, 27.7)	27.6 (12.8, 30.9)	0.440	8.6 (4.3, 11.9)	33.2 (15.9, 43.3)	0.002
IL-2	45.8 (27.8, 61.5)	47.1 (36.4, 56.6)	0.953	31.4 (21.2, 39.8)	54.9 (41.6, 65.1)	0.002
IL-21	13.5 (8.6, 15.9)	14.5 (13.2, 19.0)	0.206	3.2 (2.1, 7.4)	13.7 (10.8, 19.9)	<0.001
GF-β	1.6 (0.2, 3.6)	2.3 (1.2, 5.1)	0.513	−0.1 (−0.3, 0.1)	0 (−0.3, 0.3)	0.650
IL-17	0.3 (−0.1, 3.0)	3.6 (3.4, 4.1)	0.206	0.8 (−0.3, 1.2)	4.9 (0.5, 7.8)	0.068
**CD8+ T cells**						
TNF-α	18.4 (8.8, 21.8)	42.6 (12.4, 51.8)	0.206	10.6 (−2.8, 23.2)	51.7 (36.4, 57.1)	<0.001
IFN-γ	23.8 (8.3, 40.9)	64.2 (34.4, 69.3)	0.040	20.45 (9.1, 33.4)	60.6 (56.0, 68.0)	0.004
IL-2	19.0 (8.8, 24.5)	18.4 (11.1, 27.4)	1.000	7.3 (0.5, 9.7)	21.6 (17.0, 28.9)	<0.001
IL-21	2.4 (1.3, 4.7)	3.2 (2.6, 3.8)	0.513	0.5 (0.3, 2.2)	3.1 (1.7, 3.9)	0.010
TGF-β	0.9 (0, 1.0)	0.2 (0, 1.0)	0.679	0 (−0.1, 0.1)	−0.1 (−0.2, 0)	0.261
IL-17	0.8 (0.6, 1.9)	1.1 (1.0, 3.7)	0.055	0 (−0.1, 0.1)	0.8 (0.2, 1.2)	0.004
**NK cells**						
TNF-α	17.9 (8.8, 27.1)	43.4 (2.4, 48.4)	0.513	4.6 (1.9, 10.2)	24.1 (16.8, 40.2)	0.028
IFN-γ	42.9 (13.4, 58.8)	52.7 (7.7, 57.6)	1.000	1.9 (−2.6, 2.6)	43.9 (13.9, 59.0)	0.010
IL-2	0.4 (0.1, 0.7)	0 (0, 5.1)	0.371	−0.1 (−0.2, 0)	0 (0, 0.6)	0.128
IL-21	−0.1 (−0.3, 0.1)	0 (−0.2, 1.7)	0.310	0 (−0.2, 0.1)	−0.1 (−0.3, 0)	0.142
TGF-β	−0.1 (−0.1, 0.1)	−0.1 (−0.2, 0)	1.000	0 (−0.1, 0)	0 (−0.2, 2.4)	0.837
IL-17	0 (0, 0.3)	0 (−0.1, 0)	0.310	0 (−0.1, 0.1)	0 (0, 0)	0.711

Data compared to the baseline and presented in median values (interquartile range).

**Table 4 vaccines-10-01550-t004:** Generalized estimating equations (GEE) for neutralizing antibodies elicited by COVID-19 vaccines.

	β-Coefficient Estimate	SE	*p*
mRNA-1273 (reference category ChAdOx1-S)	29.2	9.5	0.002
Total memory cells (% B cells)	3.1	0.8	<0.001
Non-switched memory cells (% B cells)	6.4	2.7	0.016
Regulatory T (Treg) cells (% T cells)	4.2	3.6	0.242
Naïve Treg cells (% Treg cells)	−2.0	0.5	<0.001
Memory Treg cells (% Treg cells)	2.0	0.5	<0.001

All GEE models included the covariants of age, sex, WBCs and lymphocytes (%). SE, standard error.

## Data Availability

The data presented in this study are available in the article.

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
