# Peer review of "Lymphocyte Subpopulations Associated with Neutralizing Antibody Levels of SARS-CoV-2 for COVID-19 Vaccination"

_vaccines, 2022, doi:10.3390/vaccines10091550_

Round 1

Reviewer 1 Report

1. The manuscript is related to lymphocyte populations, including CD4, CD8, Treg T lymphocytes and B lymphocytes, associated with the levels of neutralizing anti-SARS Cov-2 antibodies in medical personnel vaccinated with ChAdOx1-S or mRNA-1273. The manuscript is well designed and the assays are adequate to demonstrate its hypothesis.

2. However, the manuscript has few references (31), and does not compare its results with others. There is a large number of studies that address the same topic; however, the authors generally discuss their findings in relation to other studies. Therefore, the authors must contrast their findings in a broader way in the discussion; for example, they should further discuss the decline in the population of Tregs cell.

3. The authors should contrast their results with those obtained in participants vaccinated with BNT162b1 (Sahin et al. COVID-19 vaccine BNT162b1 elicits human antibody and TH1 T cell responses. Nature. 2020 Oct;586(7830):594-599).

4. The authors should contrast their results with those described by Sokal et al. (Sokal et al. mRNA vaccination of naive and COVID-19-recovered individuals elicits potent memory B cells that recognize SARS-CoV-2 variants. Immunity. 2021 Dec 14;54(12):2893-2907.e5)

5. The authors should discuss the findings of Painter et al and compare them with their own, particularly the predominance of Th1 cells.

6. The authors should discuss the findings of Painter et al, particularly the findings of cTfh population (Painter et al. Rapid induction of antigen-specific CD4+ T cells is associated with coordinated humoral and cellular immunity to SARS-CoV-2 mRNA vaccination. Immunity. 2021 Sep 14;54(9):2133-2142.e3) .

Author Response

1. The manuscript has few references (31), and does not compare its results with others. There is a large number of studies that address the same topic; however, the authors generally discuss their findings in relation to other studies. Therefore, the authors must contrast their findings in a broader way in the discussion; for example, they should further discuss the decline in the population of Treg cells.

Response: We appreciate the suggestion very much. To contrast our findings in a broader way, additional discussion has been added. The vaccine-induced CD4+ T cells predominately exhibiting phenotypes of T helper 1 and Tfh and correlating with humoral and cellular responses; mRNA vaccination inducing robust T helper type 1 immune responses and eliciting RBD-specific memory B cells (MBCs) with high affinity toward SARS-Cov-2 variants were described. The importance of Treg cells in relation to COVID-19 and vaccination is not clear. Although the decline in the population of Treg cells, our data showed the proportion of memory Treg cells was increased and positively associated with the magnitude of functional neutralizing antibodies (nAbs). These findings indicate potential activation of Treg cells over COVID-19 vaccination might reduce the response of cytotoxic Tfh cells. However, what role of Treg cells in the effective production of nAbs is yet to be answered. Further studies are required to clarify the relationship. We have revised the manuscript (pages 10-11).

2. The authors should contrast their results with those obtained in participants vaccinated with BNT162b1 (Sahin et al. COVID-19 vaccine BNT162b1 elicits human antibody and TH1 T cell responses. Nature. 2020 Oct;586(7830):594-599) 

Response: We have contrasted the results with those obtained in participants vaccinated with BNT162b1. Our data are consistent with Sahin et al. study that demonstrated mRNA vaccination inducing robust T helper type 1 immune responses (page 11).

3. The authors should contrast their results with those described by Sokal et al. (Sokal et al. mRNA vaccination of naive and COVID-19-recovered individuals elicits potent memory B cells that recognize SARS-CoV-2 variants. Immunity. 2021 Dec 14;54(12):2893-2907.e5)

Response: We have contrasted the results with those described by Sokal et al., assessing not only the quantity but the quality of the memory B cell response. The study demonstrated mRNA vaccination effectively elicited RBD-specific MBCs with high affinity toward SARS-Cov-2 variants. Similarly, alterations of total MBCs and non-switched MBCs positively correlated with levels of nAb were clarified in our study. We have revised the manuscript (pages 10-11).

4. The authors should discuss the findings of Painter et al and compare them with their own, particularly the predominance of Th1 cells.

Response: Painter et al. study showed vaccine-induced CD4+ T cells predominately exhibiting phenotypes of T helper 1 and Tfh cells and correlating with humoral and cellular responses. We have added the information to the manuscript (page 11). However, it is hard to compare the findings of Painter et al. with our own because no markers were used to detect subpopulations of T helper 1 and Tfh cells. Besides, the evaluation of memory T cells subset distribution was not focused on vaccine-induced CD4+ T cells in the present study.

5. The authors should discuss the findings of Painter et al, particularly the findings of cTfh population (Painter et al. Rapid induction of antigen-specific CD4+ T cells is associated with coordinated humoral and cellular immunity to SARS-CoV-2 mRNA vaccination. Immunity. 2021 Sep 14;54(9):2133-2142.e3)

Response: We have added findings of Painter et al to the manuscript (page 11).  

Reviewer 2 Report

The article titled  Lymphocyte Subpopulations Associated with Neutralizing An-2 tibody Levels of SARS-CoV-2 for COVID-19 Vaccination 3 after consideration of the following minor comments.

.

1)      Abstract, depending on what , authors determine sample size.

2)      Abstract, authors should not use we i , us etc.

3)      Depending on what , authors determine sample size..

4)      Abbreviations should be mentioned when first appear for example 6-OHDA, SODI ETC.

5)      The main aim and outcome of this study is not clear.

6)      Conclusion should be improved

Author Response

1. Abstract, depending on what, authors determine sample size.

Response: The present study explored the kinetic changes of lymphocyte subsets and cytokines according to the alterations of 41 markers analyzed by 8-color flow cytometry. Due to budget limitations, 20 healthy medical staff were enrolled. Each of them had repeated measurement data for analysis.

2. Abstract, authors should not use we i , us etc

Response: Thank you for your suggestion. We have revised the abstract (page 1). 

3. Depending on what , authors determine sample size..

Response: Due to budget limitations, 20 healthy medical staff were enrolled. The Kendall's coefficient values ranged from 0.2~0.7. The overall agreement between the measures is moderate.

4. Abbreviations should be mentioned when first appear for example 6-OHDA, SODI ETC.

Response: Thank you for your suggestion. We have revised the manuscript.

5. The main aim and outcome of this study is not clear.

Response: We have clarified the main aim and outcome in the revised manuscript (page 2).

6. Conclusion should be improved

Response: We have revised the conclusion (page 11).

Reviewer 3 Report

Interesting article and prepared at a high level. Both in terms of content and editorial, the authors did a very good job. Moreover, the study provides important conclusions and there are only a few similar reports in the literature. The article has great potential as an original research worth citing.

Author Response

Thank you very much.

Reviewer 4 Report

Report vaccines-1896852

This paper deals with an important topic which is presently object of an intense research activity which has not been able, until now, to provide theoretical interpretations on the decay in time of the immune defense ability after vaccination.

This paper provides several empirical data on this matter by investigating the complex dynamics of the various populations interacting in-host. The authors correctly describe the perspective that their paper might contribute to tackle such problem. Correctly, it is expressed as a wish rather than as an effective result.

As a mathematician, I have appreciate the overall amount of information which contribute also to mathematical models which go beyond the simplistic description of deterministic population dynamics, say SIR, SEIR and various technical derivation. Indeed, the key feature of the mathematical approach to modeling  consists in describing the multiscale feature of the activation of the several components of the immune system  to contrast the invasion of virus particles.

Therefore, I strongly recommend publication of this paper, but I express the suggestion to mention on the last section also the contribution to modeling approaches. Some of them  effectively focus on the immune competition.

Author Response

I strongly recommend publication of this paper, but I express the suggestion to mention on the last section also the contribution to modeling approaches. Some of them effectively focus on the immune competition.

Response: We appreciate the suggestion very much. The contribution to modeling approaches has been added on the last section (page 10).